# Hit-to-Lead Short Peptides against Dengue Type 2 Envelope Protein: Computational and Experimental Investigations

**DOI:** 10.3390/molecules27103233

**Published:** 2022-05-18

**Authors:** Norburhanuddin Johari Zaidi, Adib Afandi Abdullah, Choon Han Heh, Chun-Hung Lin, Rozana Othman, Abdullah Al Hadi Ahmad Fuaad

**Affiliations:** 1Peptide Laboratory, Department of Chemistry, Faculty of Science, Universiti Malaya, Kuala Lumpur 50603, Malaysia; s2005724@siswa.um.edu.my; 2Drug Design & Development Research Group, Department of Chemistry, Faculty of Science, Universiti Malaya, Kuala Lumpur 50603, Malaysia; afandi_adib@live.com (A.A.A.); silverbot@um.edu.my (C.H.H.); 3Centre for Natural Products Research and Drug Discovery (CENAR), Universiti Malaya, Kuala Lumpur 50603, Malaysia; 4Department of Pharmaceutical Chemistry, Faculty of Pharmacy, Universiti Malaya, Kuala Lumpur 50603, Malaysia; 5Institute of Biological Chemistry, Academia Sinica, Nankang, Taipei 115, Taiwan; chunhung@gate.sinica.edu.tw

**Keywords:** peptide, dengue, envelope protein, molecular docking, molecular dynamics, plaque assay

## Abstract

Data from the World Health Organisation show that the global incidence of dengue infection has risen drastically, with an estimated 400 million cases of dengue infection occurring annually. Despite this worrying trend, there is still no therapeutic treatment available. Herein, we investigated short peptide fragments with a varying total number of amino acid residues (peptide fragments) from previously reported dengue virus type 2 (DENV2) peptide-based inhibitors, DN58wt (GDSYIIIGVEPGQLKENWFKKGSSIGQMF), DN58opt (TWWCFYFCRRHHPFWFFYRHN), DS36wt (LITVNPIVTEKDSPVNIEAE), and DS36opt (RHWEQFYFRRRERKFWLFFW), aided by in silico approaches: peptide–protein molecular docking and 100 ns of molecular dynamics (MD) simulation via molecular mechanics using Poisson–Boltzmann surface area (MMPBSA) and molecular mechanics generalised Born surface area (MMGBSA) methods. A library of 11,699 peptide fragments was generated, subjected to in silico calculation, and the candidates with the excellent binding affinity and shown to be stable in the DI-DIII binding pocket of DENV2 envelope (E) protein were determined. Selected peptides were synthesised using conventional Fmoc solid-phase peptide chemistry, purified by RP-HPLC, and characterised using LCMS. In vitro studies followed, to test for the peptides’ toxicity and efficacy in inhibiting the DENV2 growth cycle. Our studies identified the electrostatic interaction (from free energy calculation) to be the driving stabilising force for the E protein–peptide interactions. Five key E protein residues were also identified that had the most interactions with the peptides: (polar) LYS36, ASN37, and ARG350, and (nonpolar) LEU351 and VAL354; these residues might play crucial roles in the effective binding interactions. One of the peptide fragments, DN58opt_8-13 (PFWFFYRH), showed the best inhibitory activity, at about 63% DENV2 plague reduction, compared with no treatment. This correlates well with the in silico studies in which the peptide possessed the lowest binding energy (−9.0 kcal/mol) and was maintained steadily within the binding pocket of DENV2 E protein during the MD simulations. This study demonstrates the use of computational studies to expand research on lead optimisation of antiviral peptides, thus explaining the inhibitory potential of the designed peptides.

## 1. Introduction

Dengue is an ancient infectious disease, and its spread has significantly spiked in recent years [1]. It was estimated that there were approximately 390 million people infected by the dengue virus (DENV), of which about 100 million displayed clinical symptoms [2,3]. In 2019, concurrent with the emergence of the coronavirus disease-19 (COVID-19) pandemic, the highest number of dengue cases were recorded [1]. The increase in the number of cases may be associated with the increased time spent at home, as data in some parts of the world suggest [2,4].

DENV belongs to the family Flaviviridae, genus Flavivirus. DENV has four clinical serotypes [3,5]. Despite numerous efforts and research to develop prophylactic and therapeutic treatments, to date, alternative types of treatment to hospital care are still lacking [1]. Dengvaxia^®^ is the only commercially available vaccine against DENV, developed by Sanofi Pasteur [6]. The effort to promote the use of Dengvaxia^®^ in hotspot countries was hampered due to several complications in the use of Dengvaxia^®^ in seronegative individuals [7]. Moreover, Dengvaxia^®^ possesses varying degrees of efficacy against different clinical DENV serotypes 1 to 4 (DENV1: 50%, DENV2: 35%, DENV3: 75%, and DENV4: 75%), and this poses concerns for some countries, such as Malaysia, due to the high prevalence of DENV2 in the country [8,9]. Therefore, the search for both prophylactic and therapeutic medication is still of importance.

Peptide-based drug inhibitors have gained traction in the past several years due to their efficacy, specificity, selectivity, and low toxicity [10]. To our knowledge, Liao and Kielian [11] were the first to report DENV inhibition by an exogeneous DENV domain III (DIII) by preventing the conformational change in the envelope (E) protein during viral–host membrane fusion. Huang et al. then studied truncated DIII peptides, and four peptides were identified as leads: DN58wt, DN58opt, DS36wt, and DS36opt, where “wt” refers to “wild type”, while “opt” refers to “optimised” [12]. The optimised peptides were designed with the goal of better fitting the active site of DIII, and the peptides were designed and evaluated with residue-specific all-atom probability discriminatory function (RAPDF) scoring function. Other than DS36wt, the other peptides bind satisfactorily to the DIII of the E protein [13]. Nevertheless, their inhibition property has yet to be tested via in vitro dengue inhibition assay or in vivo challenge experiment, thus necessitating additional studies.

In this article, we report a series of short peptide fragments with a varying total number of amino acid residues (maximum ten amino acid residues) which were designed from four lead peptides: DN58wt, DN58opt, DS36wt, and DS36opt. A peptide library was designed with the goal to investigate the minimal peptide sequence required to inhibit dengue virus replication. The binding affinity of the short-chain peptides was assessed by peptide–protein molecular docking studies (via AutoDock Vina (Vina) [14]) and 100 ns molecular dynamics simulations (MD) to calculate the binding free energy using the molecular mechanics surface area protocols. The key amino acid residues were also revealed using the Discovery Studio Visualiser 4.0 (DSV) (Dassault Systèmes, CA, USA) [15]. The binding free energy calculations, as well as system stability and flexibility, energy decomposition, and hydrogen bond analyses, were performed to differentiate and justify the binding strength of the most and least active peptides toward the DENV E protein. We selected peptide fragments, synthesised via the solid-phase peptide synthesis (SPPS) protocol, and purified and characterised them using high-performance liquid chromatography (HPLC) and liquid chromatography–mass spectrometry (LCMS), respectively. These peptides were further screened for their inhibitory potential and toxicity via plaque formation assay and 3-(4,5-dimethylthiazol-2-yl)-5-(3-carboxymethoxyphenyl)-2-(4-sulfophenyl)-2H-tetrazolium (MTS) cytotoxicity assay, respectively, and compared with the peptide leads.

## 2. Results and Discussion

### 2.1. Molecular Docking Studies

Vina [14] was used for the initial blind docking of several peptides toward the DENV2 E protein crystal structure model (PDB ID: 1OAN) [16], to identify the potential binding pocket of DENV2 E protein and its key elements. Analysis of the blind docking was then carried out, which included the binding poses and affinities that were necessary to identify a small pocket in the domain I (DI)-DIII hinge region as the potential binding site of our peptides (Figure 1). A previously reported peptide inhibitor targeting the region, DET4 (which also serves as the peptide control in this experiment), was docked using a similar procedure (see Section 3.1 and Section 3.2). The docked DET4 was set as a standard reference to identify the potential correct interaction at the ligand-binding site of DENV2 E protein. The external loop from the DIII of the envelope protein is the receptor attachment site during the entry of the virus into the host cell [17]. DET4 docking results were evaluated and inspected using the DSV based on which the binding interactions and key binding residues were identified. The binding site was considered to be a standard binding site.

The estimated binding free energy values of the top 20 peptides obtained from the docking runs are summarised in Table 1. The estimated binding free energy of the DN58opt fragments ranges from −9.0 kcal/mol to −8.0 kcal/mol, smaller values indicating stronger interactions between the peptides and the protein. Unfortunately, none of the DS36wt and DS36opt-based fragmented peptides had Vina affinity scores of less than −8 kcal/mol (all of the data are available at https://bit.ly/36lXi4b accessed on 1 May 2022). The docking results demonstrated that some DN58opt peptide fragments had a better calculated binding affinity in comparison with the full 21 residues of DN58opt and the standard DET4. The peptide fragment DN58opt_8-13 with eight amino acid residues possessed the lowest binding energy (−9.0 kcal/mol), which indicates that it had the strongest interaction with the DENV2 E protein.

### 2.2. Molecular Dynamics Studies

#### 2.2.1. System Stability

The equilibration of the simulated system was evaluated based on the regulated temperature, density, potential energy, and kinetic energy. The first frame of the MD trajectory was used as a reference. The stability of the peptide binding within the DI-DIII region was assessed by calculating the root-mean-square deviation (RMSD) of the complex trajectory and ligand-only trajectory.

The calculated RMSD values indicate the difference between the backbones of DENV2 E protein, from its initial structural conformation towards its final position (oscillation of the system) [18]. In general, small RMSD fluctuations reflect the stable state of system equilibration, while sudden oscillations denote significant conformational changes in each molecule (unstable). It is expected that the RMSD deviations produced during the simulation of the peptide-bound structures will be small for a “stable” ligand–protein interaction. The RMSD values for the C-α backbone were evaluated in a 100 ns simulation for the DN58opt fragments. Of these fragments, four peptides (DN58opt_8-11, DN58opt_8-13, DN58opt_9-9, and DN58opt_10-6) were of interest in this study and are described below.

The stability of the system for the peptide fragments is illustrated accordingly: DN58opt_8-13 (Figure 2), DN58opt_8-11 (Figure 3), DN58opt_9-9 (Figure 4), and DN58opt_10-6 (Figure 5). For peptide DN58opt_8-13, the RMSD values (averaging at 3 ± 1 Å) of the ligand and E protein did not fluctuate drastically, which means that their conformations were stable. The RMSD values of the complex were also quite stable, which indicates the relatively stable binding of the ligand in the binding site of the E protein. DN58opt_8-13, with a binding affinity of -9.0 kcal/mol, is a good candidate as an inhibitor. In contrast, for peptide DN58opt_8-11, the RMSD values of the complex increased drastically, compared with the relative value at the beginning of the simulation (from about 2 Å then sharply increased to 10 Å within 34 ns), indicating that the ligand continued to move away from the binding site of E protein, and the binding might not occur. Hence, DN58opt_8-11 with a binding affinity of −8.1 kcal/mol is not a good candidate as an inhibitor. Meanwhile, for peptides DN58opt_9-9 and DN58opt_10-6, the trend of RMSD values was quite similar to DN58opt_8-13, although the RMSD values were not as consistent as the values of DN58opt_8-13 (averaging at 4 ± 2 Å). Thus, DN58opt_9-9, with a binding affinity of −8.4 kcal/mol, and DN58opt_10-6, with a binding affinity of −8.4 kcal/mol, could also be potential peptide inhibitors.

In contrast to RMSD values that illustrate ligand–receptor deviation, the atomic root-mean-square fluctuation (RMSF) calculates residue flexibility (C_α_ fluctuation) during MD simulations (data not shown but are available at https://bit.ly/36lXi4b accessed on 1 May 2022). Overall, the DENV2 E protein (receptor) amino acid residues fluctuated between 1 and 5 Å throughout the simulation, except for when it was bound to DN58opt_6-10 and DN58opt_8-11, in which case the 390th to 400th residues fluctuation increased by about 8 Å. The RMSF range suggests the peptide complex was structurally stable (less fluctuation) within the 100 ns MD simulations and confirms the observed RMSD deviation of the ligand–receptor complex for DN58opt_8-11. 

#### 2.2.2. Calculation of Free Energy Binding

A summary of the free energy calculation for selected peptide fragments is summarised in Table 1. The reported values are the top 20 peptide fragments having the best affinity values based on Vina. From the data, it is predicted that DN58opt_10-6 would be the most potent DENV2 E protein inhibitor of this series, with molecular mechanics generalised Born surface area (MMGBSA) value of −82.5097 kcal/mol and molecular mechanics Poisson–Boltzmann surface area (MMPBSA) value of −77.5954 kcal/mol. In contrast, the peptide fragment of DN58opt_8-13, which possessed the highest Vina affinity of −9.0 kcal/mol and the most stable calculated RMSD values (3 ± 1 Å) within 100 ns simulation, had MMGBSA and MMPBSA free binding energy values of about half the values of DN58opt_10-6, at −41.8317 kcal/mol and −39.4506 kcal/mol, respectively. Although the peptide fragment of DN58opt_9-9 shared a similar Vina affinity with DN58opt_10-6 (-8.4 kcal/mol), its MMGBSA and MMPBSA values were even lower, at −40.1397 kcal/mol and −44.0312 kcal/mol, respectively. Finally, the DN58opt_8-11 peptide fragment, with −8.1 kcal/mol Vina affinity and the least stable peptide–protein binding from RMSD calculations, also had the highest MMGBSA and MMPBSA values of −0.5408 kcal/mol and −2.2644 kcal/mol.

Table 2 illustrates the detailed changes in free energy of binding (ΔE_binding_) of the peptides by averaging snapshots from MD trajectories using the MMPBSA method. The effective polarizable bond (EPB) parameters illustrate unfavourable binding total electrostatic contributions, while van der Walls (vdW) interactions, energy electrostatic (EEL), and nonpolar parts generally show favourable interactions. Overall, net favourable interactions were observed for all three peptide candidates. Data for the remaining 17 peptide fragments are available at https://bit.ly/36lXi4b accessed on 1 May 2022.

To determine the important amino acids of the DN58opt peptide fragments that contribute to the binding onto the DENV2 E protein, the energy decomposition was calculated for the DN58opt-bound complexes. Figure 6 illustrates the decomposed energies based on a per residue basis for the peptides DN58opt_10-6, DN58opt_8-13, and DN58opt_9-9 (raw energy decomposition data for the other peptides are available at https://bit.ly/36lXi4b access on 1 May 2022). The observed energy per residue correlated well with the observed RMSF values (the minimum energy value per residue also had the minimum RMSF value).

#### 2.2.3. Interaction of DN58opt Peptide Fragments in Active Sites of DENV2 E Protein

The possible binding interactions between DENV2 E protein and DN58opt peptide fragments were further analysed to explore the contributing interactions in the active sites of the designed peptides. The binding interactions were mapped into a 2D diagram using DSV and selected peptides are illustrated in Figure 7. In general, key E protein amino acids, with more than 15 out of the top 20 peptides interacting with these residues, were LYS36, ASN37, ARG350, LEU351, and VAL354 (docked ligand in PDB format and 2D receptor–ligand interactions for the top 20 peptides are accessible at https://bit.ly/36lXi4b access on 1 May 2022). We believe that these residues might play crucial roles in the effective binding interactions between the peptide fragments and the E protein. However, no further specific investigation (i.e., mutation analysis, followed by simulation studies) was instigated.

There were seven main E protein residues interacting with the DN58opt_10-6 peptide fragment, five of which are listed in this study, i.e., SER16, LYS338, ASN37, ASN355, and LEU351; these residues were involved in hydrogen bonding, while PHE11 contributed to the pi–pi (π–π) stacking interaction. Pi–sigma (π–σ) interaction was also detected at the VAL354 residue. Additionally, the analysis indicated that some vdW interactions were found between the DN58opt_10-6 peptide and the binding residues.

In addition, ILE335, GLY349, LEU351, ASN355, and LYS38 were involved in hydrogen bonding, whereas LYS36, VAL347, PRO356, VAL354, and LEU294 contributed to the alkyl and π–alkyl interactions for the DN58opt_8-13 peptide fragment. Furthermore, vdW interactions were found between the DN58opt_8-13 peptide fragment and the binding residues of PHE337, PRO336, MET297, GLU13, ASN37, MET34, ALA35, LEU348, ILE339, PRO39, and THR40.

The DN58opt_9-9 peptide fragment residues—ASN37, LEU348, PHE337, LEU351, ASN355, and GLU370—were involved in the hydrogen bonding. Additionally, the residue of ARG350 contributed to the pi–alkyl interaction. Moreover, vdW interactions were found between the DN58opt_9-9 peptide drug and the binding residues of LYS334, MET297, LYS38, SER16, GLY17, SER16, VAL347, ILE339, GLY14, GLY349, MET301, GLU338, ALA35, PRO371, PRO372, PRO350, ILE352, and MET34. The analysis of the top 20 contact residues (receptor–ligand interactions) is available at https://bit.ly/36lXi4b accessed on 1 May 2022. 

### 2.3. Summary and Selection of Lead Peptides

A total of 11,699 peptide fragments were generated from this study. The extracted data for all peptide fragments can be obtained at https://bit.ly/36lXi4b accessed on 1 May 2022. Comparative analysis between the data gathered from the in silico studies of DN58opt peptide fragments suggests DN58opt_7-12, DN58opt_8-13, and DN58opt_10-6, as the top 3 out of the top 20 potential DENV2 antiviral peptides. However, we selected two candidates—DN58opt_8-13 and DN58opt_9-9—to be synthesised and subjected to biological assays. DN58opt_8-13 was chosen, instead of DN58opt_7-12 or DN58opt_10-6, due to having the best-predicted docking value from Vina, ranking first out of the three, although it ranked last for both MMGBSA and MMPBSA studies, in comparison with DN58opt_7-12 and DN58opt10-6. Despite the predicted strong Vina affinity, as well as MMGBSA and MMPBSA energy values of DN58opt_10-6, it was not selected to be further synthesised and used in biological assay because there is a potentially unfavourable positive–positive repulsion between Lys36 residue of E protein with the histidine residue (HIS6: 6th amino acid) of DN58opt_10-6, as well as a potentially unfavourable donor–donor repulsions between ARG350 residue of E protein with the tyrosine and cysteine residues (TYR1 and CYS3: first and third amino acids, respectively) of DN58opt_10-6 (see Appendix A, at https://bit.ly/36lXi4b accessed on 1 May 2022). These repulsive interactions would pose the risk of making DN58opt_10-6 unable to remain stable at the binding site in the biological assay, especially since the predicted MD stability was only on a nanosecond scale. 

Additionally, DN58opt_8-13 had the shortest peptide sequence of eight residues, thus bringing future perspectives of shorter synthesis time and lower costs into consideration. Finally, DN58opt_9-9 served as the least performing peptide among the 20 peptides, when considering the results from both docking and dynamics simulation studies.

### 2.4. Safety and Efficacy Evaluation of Peptides against DENV2 E Protein

In vitro analysis was conducted for the synthesised peptides, DN58opt_8-13 and DN58opt_9-9, and the control peptide, DET4. The studies involved cytotoxicity assay and viral plaque formation assay.

#### 2.4.1. Cytotoxicity Assay (MTS)

The cytotoxicity of the DN58opt peptide fragments on Vero cells was determined using the MTS assay, as per manufacturer recommendation (see Section 3.4.1). The cell percentage viability (% viability) of DN58opt_8-13, DN58opt_9-9, and DET4 at different concentrations are illustrated in Figure 8. The maximum nontoxic dose (MNTD) for DN58opt_8-13 and DET4 was 400 µM, while the MNTD for DN58opt_9-9 was 200 µM. In general, the peptides showed little-to-no cytotoxicity, up to 200 µM concentration. A similar observation was previously reported for short peptides derived from DN58opt [19]. 

#### 2.4.2. Plaque Formation Assay

The synthesised peptides were screened in vitro to determine their potential inhibitory activity against DENV2 using a pretreatment plaque formation assay. This assay is commonly used to assess the virucidal activity and viral entry inhibition [19] and has previously been used as a standard inhibitory assay for dengue screening experiments [19,20]. Since there is no global peptide “standard” for DENV plaque formation assay, we selected DET4, a previously reported peptide inhibitor, as our peptide standard. Additionally, we included a nonpeptide viricidal, ribavirin, as the positive control. Although ribavirin’s mechanism of action is not via the inhibition of the DENV fusion process, we observed that ribavirin is also the positive control of choice for this type of assay [20,21,22].

DN58opt_8-13 showed significant inhibitory activity, compared with control (no peptide–“DENV DMSO”), by reducing the plaque formation at 63.42% ± 5.84% (** *p*-value < 0.01), followed by DN58opt_9-9 at 37.64% ± 0.85% (** *p*-value < 0.01) (Figure 9). Both peptides showed lower viral load (better inhibition), compared with DET4 (peptide standard, 26.57% ± 7.82%, *** *p*-value < 0.001), and ribavirin inhibited the plaque formation by 73.35% ± 9.38% (** *p*-value < 0.01). It is worth mentioning that there was no significant difference between the inhibitory data from DN58opt_8-13 and ribavirin.

The data from the biological assays of DN58opt_8-13 showed a superior inhibitory effect on DENV2 in comparison with DN58opt_9-9. This observation corroborates with the docking and MD data listed in Table 1, according to which lower energy was required for DN58opt_8-13 (−9.0 kcal/mol) to bind with the targeted E-protein than DN58opt_9-9 (−8.4 kcal/mol).

## 3. Materials and Methods

### 3.1. Molecular Docking Studies

#### 3.1.1. Model of Envelope Protein and Its Known Inhibitor

Crystal structure of DENV2 E protein was retrieved from RCSB Protein Data Bank (PDB ID: 1OAN). The structure is in a prefusion state and made up of homodimeric form. Only one monomer of the E protein dimer, chain A of the 1OAN structure, was considered for this study, as both monomers are in the same conformation, with an RMSD of 0.365 Å. The E protein structure was verified for any missing residue or atom, alternate residue conformation, and nonbonded atoms using DSV [15]. All water and bound ligands found were removed using DSV. A known peptide inhibitor targeting DIII of the E protein was identified from the literature, coded as DET4 (IC50 = 35 μM) [23]. A model of DET4, which is made up of 10 amino acid residues, AGVKDGKLDF, was built using UCSF Chimera software version 1.11.2 (Resource for Biocomputing, Visualization, and Informatics, CA, USA) [24,25].

#### 3.1.2. Construction of Short-Chain Peptide Library

A library of short-chain peptides comprising 6 to 10 amino acid residue was constructed based on the fragmentation of DN58opt, DS36wt, and DS36opt peptides. An in-house python script utilising the UCSF Chimera peptide builder function [25] and minimisation function was used to automatically build the peptide model of 6–10 residue-sliding windows (Figure 10). Dunbrack rotamer library [26] was used for the determination of side-chain conformations. First, phi and psi angles were assigned randomly between −180° and 180° using Python. Then, the built peptides were subjected to two stages of minimisations with Amber ff14SB force field [27]: 1000-step steepest descent and 1000-step conjugate gradient.

#### 3.1.3. Virtual Screening of the Peptide Library via Molecular Docking

Virtual screening of the peptide library was carried out by employing the molecular docking method using AutoDock Vina software version 1.1.2 (The Scripps Research Institute, San Diego, CA, USA) [14]. Polar hydrogen atoms were added to the receptor (E protein) and the ligands (peptide library) to ensure hydrogen bond formation/detection during scoring function calculation, while nonpolar hydrogens were merged. The E protein was set to be fully rigid. In contrast, the peptides were set to be fully flexible except for the amide bond (default setting), as it is highly energy unfavourable to be rotated [28]. This procedure was completed using AutoDockTools (ADT) included in MGLTools 1.5.6. For the batch operation of the peptide library, an ADT script—namely, “prepare_ligand4.py”—was used via a command-line interface (CLI). The results were saved in pdbqt format. 

A blind docking approach was employed based on which the search space was set to be the whole surface of the E protein. A grid box with 1 Å grid spacing for the search space was visualised and identified via ADT. The grid box volume was 54 × 74 × 130 Å^3^, and its centre was at the x, y, and z coordinates −11.500, 75.048, and 37.183, respectively. The grid box information was saved in a text file named “conf.txt” following Vina input format specification.

Since the Vina docking approach is stochastic, the exhaustiveness setting was increased to 100, and the docking procedure was repeated ten times with random seeding [14,29]. This is essential to cater to the high torsion number of the peptides. The virtual screening process was executed using a bash script via CLI.

The standard peptide inhibitor, DET4, was docked into the E protein following the same procedure. As information regarding the dengue E protein–DET4 complex crystal structure is not available, and no reported raw computational data were available at hand for comparison, our docked DET4 herein was set as a standard reference for the potential correct interaction at the ligand-binding site of dengue E protein.

#### 3.1.4. Analyses of the Virtual Screening Results

DET4 docking result was evaluated and inspected using DSV, based on which binding interactions and key binding residues were identified. The 10-time repeated virtual screening results were pooled and sorted based on Vina binding affinity, followed by the percentage of binding at the standard binding site. The process was performed using a bash script via CLI.

### 3.2. Molecular Dynamics Studies

#### 3.2.1. Molecular Dynamics Simulation of the E Protein–Peptide Complexes

The top 20 peptides (based on the best average score of binding affinity—Table 1) from the virtual screening in complex with E protein were subjected to 100 ns MD simulation using Assisted Model Building and Energy Refinement (AMBER) program version 14 [30]. The E protein bound with each peptide was parameterised with the Amber ff99SB-ILDN force field [31], which was developed specifically for protein simulation. The protonation state of the complex was based on a neutral pH condition. The complex was solvated with a triangle 3-point (TIP3P) water model in a rectangular box with a 12 Å margin. The TIP3P water model was chosen as its parameterisation development was the same as the Amber force field. Sodium and chloride ions were added when necessary to neutralise the system.

The simulations were performed in a periodic boundary condition (PBC); the 12 Å margin water box prevented the complex from overlapping its mirror image when it crossed the boundary. The particle mesh Ewald (PME) method was used to calculate electrostatic interactions, with a cutoff of 8.0 Å to limit direct space sum calculation. The neutralised system with appropriate counterion was critical to prevent the electrostatic charge from going to infinity due to the imposed PBC [32]. The vdW interactions were truncated at 8.0 Å as they generally decay significantly after 8 to 10 Å distance due to the 1/r6 dependence [33]. The simulation was run with two-femtosecond (fs) time steps while constraining heavy atom hydrogen bonds using the SHAKE/SETTLE algorithm [34,35]. The temperature was regulated at 300 K using Langevin dynamics, with a collision frequency of 2.0 ps^−1^. The pressure was regulated where necessary at 1 atm using Berendsen barostat.

Initially, two-stage short minimisation was run to remove wrong contacts and relax the system. The first stage was minimising the water model while restraining the complex, followed by minimising all components in the system in the second stage. For each minimisation stage, 500 steps of steepest descent, followed by 500 steps of conjugate gradient algorithms, were employed. While restraining the complex, the minimised system was gradually heated up from 0 K to 300 K in a 50 ps canonical ensemble (NVT), followed by density equilibration in a 50 ps isothermal–isobaric ensemble (NPT). Finally, the systems were run in an NPT ensemble for 500 ps to gauge the equilibration of the system and extended to 100 ns for further evaluation. All simulations were run using the GPU version of the Particle Mesh Ewald Molecular Dynamics (pmemd.cuda) module in AMBER 14 software. 

#### 3.2.2. Analyses of the Molecular Dynamic Trajectories

The equilibration of the simulated system was assessed based on the regulated temperature, density, potential energy, and kinetic energy. Using the first frame of the MD trajectory as a reference, the RMSD of the complex trajectory and ligand-only trajectory were calculated to assess the stability of the binding. Pairwise RMSD between each frame of the ligand-only trajectory was also calculated. The RMSF of each residue of the complex was also calculated to confirm the stability further. In addition, hydrogen bonding interactions between the protein and peptide were simulated by plotting the 2D diagram showing the binding interactions by using PoseView.

#### 3.2.3. Calculation of Free Energy of Binding

The simulated complexes’ free energy of binding was calculated using molecular mechanics energies combined with the generalised Born or Poisson–Boltzmann and surface area continuum solvation (i.e., MMGBSA and MMPBSA, respectively). The calculation was performed on 400 frames throughout 100 ns simulation time using the MMPBSA.py script included in AMBER Tools 16 software [36]. In addition, MMGBSA energy decomposition was also calculated using the same script in the same operation (MMPBSA.py input file is included in the Appendix A) to identify critical residues of both protein and peptide that contribute to the free energy of binding. The identification of the key residues is essential to further optimise the peptide lead in the future.

### 3.3. Peptide Synthesis

All reagents were purchased for Merck (Germany) unless otherwise mentioned. The peptides (0.05 mmol) were synthesised following the conventional Fmoc solid-phase peptide synthesis approach, as previously reported [20,37]. Generally, the synthesis involves the coupling of the first amino acid to the SPPS resin (Rink Amide MBHA, 100-200 mesh, Sigma-Aldrich). A repeated cycle of N-terminal Fmoc removal using 2-methylpiperidine and double coupling of preactivated amino acids (HCTU and DIPEA) were carried out until sequence completion. Finally, peptides were removed from the solid support using the TFA–TIS–water (95:2.5:2.5% *v*/*v*) cocktail. The peptides were precipitated out using cold diethylether before being resuspended in 50% *v*/*v* acetonitrile–water for lyophilisation.

Peptide crudes were purified via reverse-phase HPLC at room temperature, using Atlantis C18 column (5 um, 4.6 × 100 mm) at a gradient of 2–90% ACN/H_2_O (0.1% TFA) over 30 min, with a flow rate of 1 mL/min. The eluent was characterised by monitoring the wavelength at 214 nm. The final products were verified by HPLC and HR-QQQ-LCMS. DN58opt_8-13; Yield: 14 mg (23.35%). Molecular weight: 1199.37 g/mol. ESI-MS; [M + H^+^]^+^
*m*/*z* 1200.24 (calcd. 1200.37), [M + 2H^+^]^2+^ *m*/*z* 600.15 (calcd. 600.68). RP-HPLC; R_t_. 14.8 min, purity ≥95%. DN58opt_9-9; Yield: 13 mg (19.56%). Molecular weight: 1329.53 g/mol. ESI-MS; [M + 1H^+^]^1+^ *m*/*z* 1329.68 (calcd. 1330.53). RP-HPLC; R_t_ 17.1 min, purity ≥95%. DET4; Yield: 16 mg (30.50%). Molecular weight: 1049.18 g/mol. ESI-MS; [M + 1H^+^]^1+^ *m*/*z* 1049.56 (calcd. 1050.18). RP-HPLC; R_t_ 13.3 min, purity ≥95%

### 3.4. Safety and Efficacy Evaluation of Peptides against DENV2 E Protein

#### 3.4.1. MTS Cytotoxicity Assay

Vero cells were seeded in a 96-well plate and incubated overnight. The peptides were prepared in six different concentrations (0, 25, 50, 100, 200, and 400 µM) in DMEM culture medium. The media in the wells were replaced with fresh media containing the peptides or DMSO as a control. After 48 h, cell viability was assessed using CellTiter 96^®^ AQ_ueous_ Nonradioactive Cell Proliferation assay (Promega), according to the manufacturer’s instructions (https://bit.ly/promegaMTS, accessed on 1 January 2021). The absorbance was recorded using a Bio-Rad Model 680 Microplate Reader (Hercules, CA, USA). The absorbance for each well was corrected against a no-cell blank and normalised to their respective DMSO control wells. Each sample was assayed in duplicates, and the experiment was repeated three times (*n* = 3).

#### 3.4.2. Plaque Formation Assay

Vero cells were plated in 12-well plates and incubated for 24 h at 37 °C with 5% CO_2_. A 150 PFU of DENV2 was incubated with 100 mM of the peptides or ribavirin at 37 °C for 30 min. DMSO (1% *v*/*v*) and ribavirin served as the negative and positive controls, respectively. The peptide and virus mixtures were then added to the cells. Infection and plaque assays were carried out as previously mentioned [20,23]. The number of the plaques in the treated samples was counted and plotted as a percentage of plaque formation normalised to the negative control to calculate the inhibition percentage. The assay was repeated at least three times.
$$Inhibition\;\left(\%\right)=100\%-normalized\;plaque\;formation\;(\%)$$

## 4. Conclusions

In conclusion, we designed short new peptide fragment hits (6–10 amino acid residues) from the lead peptides—DN58wt, DN58opt, DS36wt, and DS36opt. By leveraging molecular docking (AutoDock Vina) and MD simulations, a library of 11,699 compounds was generated and studied, and the top 20 peptide fragments were reported in this study. Two peptide fragments were selected (DN58opt_8-13 and DN58opt_9-9, which were the best and worst of the 20 fragments, respectively) and synthesised via SPPS. In silico Vina calculation results showed DN58opt_8-13 to have the lowest binding energy score of −9.0 kcal/mol, while MD simulation demonstrated that DN58opt_8-13 possessed strong protein–ligand interactions at the DENV DI-DIII binding site and remained within the cavity throughout the 100 ns MD simulations. Key E protein residues involved in the binding of the top 20 peptide fragments were LYS36 (19 out of 20 peptides), ASN37 (15 out of 20 peptides), ARG350 (15 out of 20 peptides), LEU351 (19 out of 20 peptides) and VAL 354 (20 out of 20 peptides)—a balance between polar and nonpolar residues. The DN58opt_8-13 peptide also showed the best inhibitory activity against DENV2 (inhibition % = 63.42% ± 5.84% at 200 μM) in contrast to the other tested peptides. Overall, the in silico and in vitro results showed excellent correlation, indicating that DN58opt_8-13 has the potential to serve as the starting point for lead optimisation and development of future anti-dengue compounds.

In the future, our hit-to-lead sub-automation approach would be promising for the development of inhibitors against other Flaviviruses such as West Nile, tick-borne encephalitis, yellow fever, Zika, etc. since the ligand library was generated in this study, and the batch script was written. We are also looking into integrating machine learning or artificial intelligence approaches to further help us in developing suitable inhibitors that will hopefully be able to combat these diseases.

## Figures and Tables

**Figure 1 molecules-27-03233-f001:**
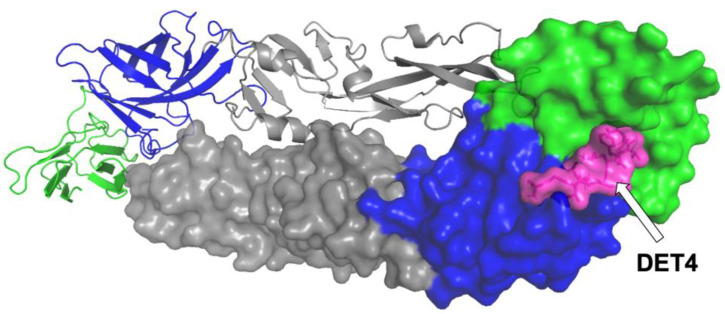
Binding pocket of DENV2 E protein. DET4 (in magenta) binds at the DI-DIII hinge (DI in green and DIII in blue) is illustrated here. Coloured figure is available online.

**Figure 2 molecules-27-03233-f002:**
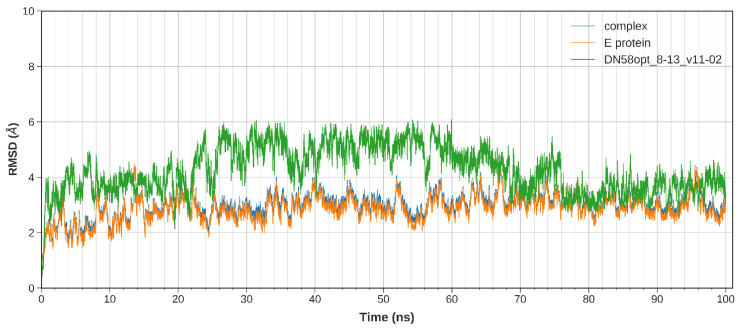
Plot of 100 ns MD simulation for the DN580pt_8-13 peptide fragment. Slight overlapping between the complex data (blue) and the E protein data (orange). Coloured figure is available online.

**Figure 3 molecules-27-03233-f003:**
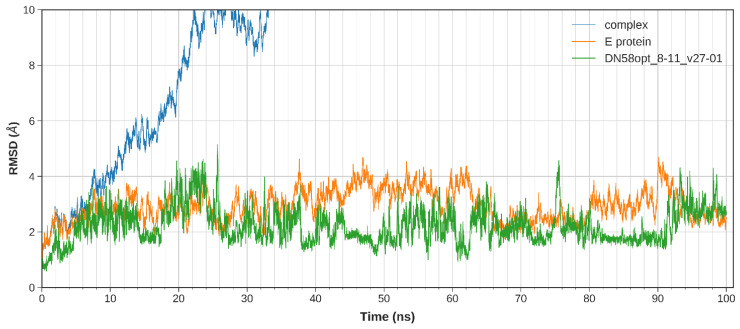
Plot of 100 ns MD simulation for the DN580pt_8-11 peptide fragment. The peptide was stable for the first 10 ns and then quickly destabilized and moved out of the pocket by 35 ns simulation. Coloured figure is available online.

**Figure 4 molecules-27-03233-f004:**
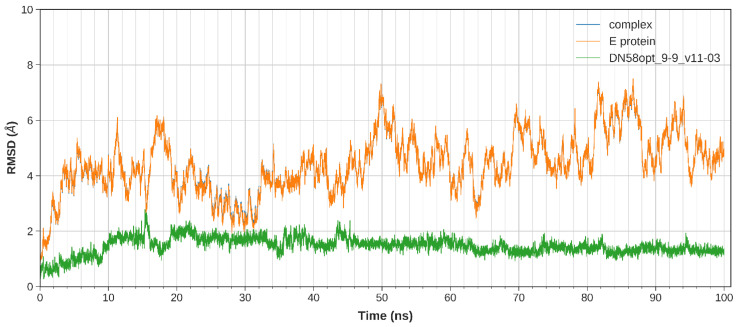
Plot of 100 ns MD simulation for the DN580pt_9-9 peptide fragment. The complex data (blue) overlap with the E protein data (orange) throughout the simulation. Coloured figure is available online.

**Figure 5 molecules-27-03233-f005:**
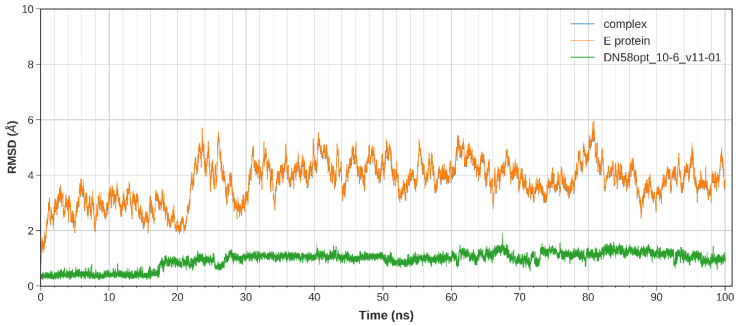
Plot of 100 ns MD simulation for the DN580pt_10-6 peptide fragment. The complex data (blue) overlap with the E protein data (orange). Coloured figure is available online.

**Figure 6 molecules-27-03233-f006:**
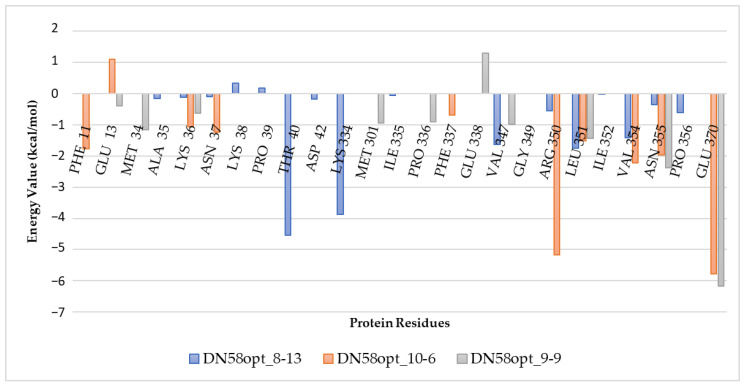
Combined free energy decomposition per residue of DN58opt_8-13, DN58opt_10-6, and DN58opt_9-9. The positive and negative values indicate the unfavourable and favourable contributions for the binding, respectively.

**Figure 7 molecules-27-03233-f007:**
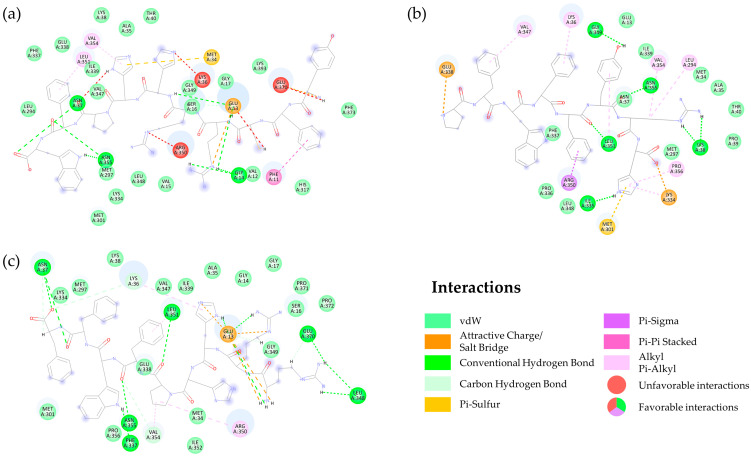
Schematic representation of the binding interactions between the DENV2 E protein (circled residues—in green/orange/pink/red/purple) and the peptide fragments (line representation): (**a**) DN58opt_10-6; (**b**) DN58opt_8-13; (**c**) DN58opt_9-9, peptide fragments. Coloured image is available online.

**Figure 8 molecules-27-03233-f008:**
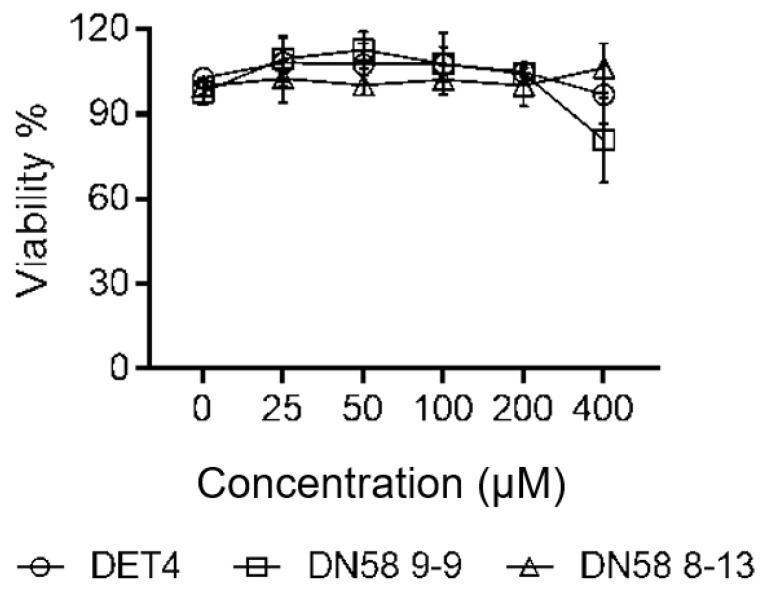
Vero cells viability (in percentage viability, Viability %) following 48 h exposure to DN58opt_8-13, DN58opt_9-9, and DET4, at various concentrations (0–400 µM).

**Figure 9 molecules-27-03233-f009:**
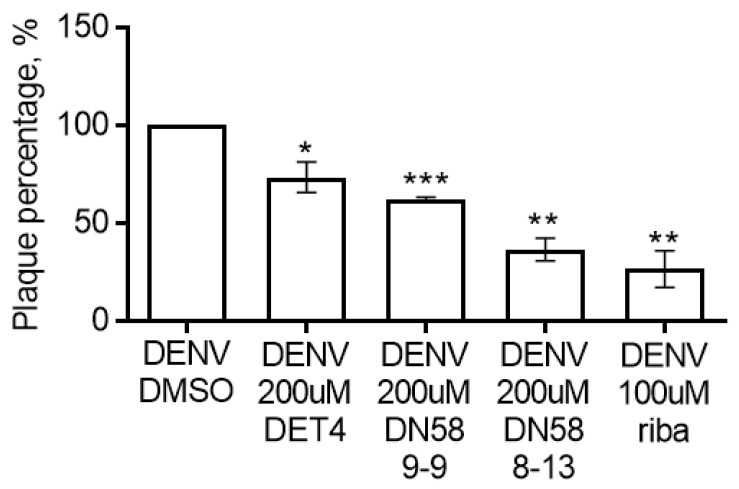
Antiviral activity of synthesised short peptide fragments against DENV2 using plaque formation assay. * *p* < 0.05, ** *p* < 0.01, *** *p* < 0.001.

**Figure 10 molecules-27-03233-f010:**
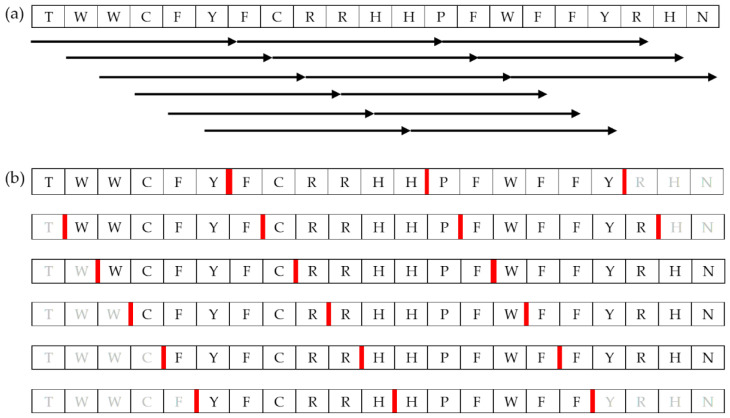
Illustration of six-residue peptide fragmentation of DN58opt as an example. The same procedure was applied to 7-, 8-, 9- and 10-amino-acid peptides: (**a**) sliding windows of 6 amino acids, with 1 amino acid step size; (**b**) a different illustration of the sliding windows. Greyed residues are not a 6-amino-acid fragment; thus, it was rejected. The bold red line is where the peptide was cut.

**Table 1 molecules-27-03233-t001:** Top 20 peptides with their Vina affinity and free energy of binding in kcal/mol. Peptides were sorted according to descending affinity values (excluding DET4).

Peptide	Sequence	Vina Affinity (kcal/mol)	MMGBSA (kcal/mol)	MMPBSA (kcal/mol)
DET4 (standard)	AGVKDGKLDF	−6.6	−41.3249	−40.9460
**DN58opt_8-13**	PFWFFYRH	−9.0	−41.8317	−39.4506
DN58opt_7-12	HHPFWFFY	−8.7	−54.4044	−44.3665
DN58opt_6-1	TWWCFY	−8.6	−29.4070	−27.2727
DN58opt_6-11	HHPFWF	−8.6	−23.2421	−23.3695
DN58opt_7-10	RRHHPFWF	−8.6	−35.1696	−31.8476
DN58opt_6-10	RHHPFW	−8.4	−11.9393	−16.1549
DN58opt_6-13	PFWFFY	−8.4	−35.0189	−27.4746
DN58opt_7-9	CRRHHPFW	−8.4	−37.2646	−38.2679
**DN58opt_9-9**	RRHHPFWFF	−8.4	−40.1397	−44.0312
**DN58opt_10-6**	YFCRRHHPFW	−8.4	−82.5097	−77.5954
DN58opt_6-2	WWCFYF	−8.3	−39.4207	−31.8778
DN58opt_6-12	HPFWFF	−8.3	−34.4456	−32.1598
DN58opt_6-15	WFFYRH	−8.3	−38.8074	−36.8260
DN58opt_9-11	HHPFWFFYR	−8.2	−52.0706	−46.9053
DN58opt_8-2	WWCFYFCR	−8.1	−48.6330	−44.6472
**DN58opt_8-11**	HHPFWFFY	−8.1	−0.5408	−2.2644
DN58opt_9-1	TWWCFYFCR	−8.1	−20.8016	−20.7366
DN58opt_6-16	FFYRHN	−8.0	−17.8920	−20.2634
DN58opt_7-2	WWCFYFC	−8.0	−36.8728	−31.1389
DN58opt_7-11	RHHPFWFF	−8.0	−29.0445	−24.6885

**Table 2 molecules-27-03233-t002:** Relative binding free energies of complexes estimated using MMPBSA.

Peptide	vdW	EEL	EPB	EN POLAR	ΔE_binding_ (kcal/mol)
DN58opt_8-13	−50.4578 ± 0.3663	−146.8887 ± 1.2591	163.4684 ± 1.2054	−5.5725 ± 0.0259	−39.4506 ± 0.4436
DN58opt_10-6	−93.4038 ± 0.3287	−457.2968 ± 1.9056	482.7020 ± 1.7063	−9.5968 ± 0.0205	−77.5954 ± 0.5297
DN58opt_9-9	−58.7485 ± 0.3692	−358.8480 ± 1.5831	381.1478 ± 1.4254	−7.5825 ± 0.0244	−44.0312 ± 0.5104

Note: vdW and EEL represent the van der Waals and electrostatic contributions from MMPBSA, respectively. Contribution to the solvation free energy (EPB), and EN POLAR correspond to the nonpolar contribution to the solvation free energy. ΔE binding is the final estimated binding free energy.

## Data Availability

Selected data (i.e., AutoDock Vina and MD output, PDB file format, 2D receptor–ligand binding interactions, and the energy decomposition) for the top 20 peptides are available in Google Drive (Domain: Universiti Malaya) and are publicly accessible (view/download only) at https://bit.ly/36lXi4b from 1 May 2022 until May 2025.

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
