# Peer review of "Hit-to-Lead Short Peptides against Dengue Type 2 Envelope Protein: Computational and Experimental Investigations"

_molecules, 2022, doi:10.3390/molecules27103233_

Round 1

Reviewer 1 Report

  1. The title is interesting and well written by the authors but still, lacks some issues like the abstract section is not written well. The authors are advised to add more information that will allow a reader to understand the key findings of the manuscript.
  2. The introduction section is poorly written and it constitutes a lack of crux in it.
  3. The authors are advised to mention the full form first later on they are allowed to add abbreviations.
  4. In line 109 write the full form of DSV and mention its reference.
  5. Figure 1 shows the complete structure of 1OAN with ligand DET4. Label the diagram.
  6. The sentence “In general, key E protein amino acid residues, having had more than 15/20 peptides interacting, were LYS36, 222 ASN37, ARG350, LEU351, and VAL354 (docked ligand in PDB format and 2D receptor-ligand interactions for the top 20 peptides are accessible at https://bit.ly/36lXi4b).” line 221 is not clear. Are the interactions with these residues common among other 20 peptides and 1OA3?
  7. Explain the significance of interactions observed in the complexes. Interaction analysis could be performed after docking and then it could be further assessed by simulation studies.
  8. The manuscript needs standard review for enriched referencing, especially in the introduction section. For example, from lines 46 to 56 only a single citation is there.
  9. Please do add the future perspective of this study.

Author Response

Reply to Reviewers

Dear reviewers,

I thank you for the feedback to better improve our manuscript. Please find our answers below. All referencing (Page number and Lines) is referring to the latest manuscript with track changes.

Thank you.

Yours Sincerely,

Abdullah Al-Hadi Ahmad Fuaad

Corresponding Author 1

Comments and Suggestions for Authors from Reviewer 1

  1. The title is interesting and well written by the authors but still, lacks some issues like the abstract section is not written well. The authors are advised to add more information that will allow a reader to understand the key findings of the manuscript.

We thank you for the suggestion.

The abstract was adjusted to remove some less critical information and key findings were added.

 From Line 19: “Although promising, the only commercially available vaccine, Dengvaxia®, is hampered with the issue of eliciting autoimmune responses to dengue seronegative individuals. Extensive research is under development in exploring dengue antiviral peptide drugs.” was removed.

 The sentence, “via Molecular Mechanics-Poisson Boltzmann Surface Area (MMPBSA) and Molecular Mechanics Generalized Born Surface Area (MMGBSA) methods.” was added to further iterate the specific method used in this study.

 And a short explanation, “Our studies identified that the electrostatic interaction (from free energy calculation) to be the driving stabilizing force for the E protein-peptides interactions. Five key E protein residues were also identified that had the most interactions with the peptides: (Polar) LYS36, ASN37, and ARG350, and (Non-Polar) LEU351 and VAL354, and these residues might play a crucial role in the effective binding interactions. One of the peptide fragments… ” was added to add more relevant information.

  1. The introduction section is poorly written and it constitutes a lack of crux in it.

We thank you for the comment.

We have tried out best to write a content that covers a wider scope that would help newer researcher understand the views and prospects of “why” we this study was conducted. Following the comment, we have restructured the introduction – to remove what we believe could be the “crux-lacking” section and expand in some other sections. Please refer to the resubmitted manuscript for the complete rewrite.

  1. The authors are advised to mention the full form first later on they are allowed to add abbreviations.

Thank you for the advised. We have relooked at the whole document and added the followings:

Page 2 Line 60: …Coronavirus Disease-19 (COVID-19)…

Page 2 Line 71: … serotypes 1 to 4 (DENV1…)…

Page 2 Line 92: … Autodock Vina (Vina)…

Page 2 Line 100: … high performance liquid chromatography (HPLC)…

Page 2 Line 101: … liquid chromatography-mass spectrometry (LCMS)…

Page 2 Line 103: … 3-(4,5-dimethylthiazol-2-yl)-5-(3-carboxymethoxyphenyl)-2-(4-sulfophenyl)-2H-tetrazolium (MTS)…

Page 6 Line 302: … effective polarizable bond (EPB)…

Page 6 Line 303: … energy electrostatic (EEL)…

  1. In line 109 write the full form of DSV and mention its reference.

Dear reviewer, DSV was already mentioned on Page 1 Line 95. Reference was added as requested.

  1. Figure 1 shows the complete structure of 1OAN with ligand DET4. Label the diagram.

Figure is relabelled as suggested.

  1. The sentence “In general, key E protein amino acid residues, having had more than 15/20 peptides interacting, were LYS36, 222 ASN37, ARG350, LEU351, and VAL354 (docked ligand in PDB format and 2D receptor-ligand interactions for the top 20 peptides are accessible at https://bit.ly/36lXi4b).” line 221 is not clear. Are the interactions with these residues common among other 20 peptides and 1OA3?

Thank you for pointing it out. We have rephrased the sentence to make it clearer. The new sentence is as follows:

Page 7 Line 328, “In general, key E protein amino acid, having had more than 15 out of the top 20 peptides interacting with these residues, were LYS36, ASN37, ARG350, LEU351 and VAL354…”

and

Page 14 Line 627, “…fragments were LYS36 (19 out of 20 peptides), ASN37 (15 out of 20 peptides), ARG350 (15 out of 20 peptides), LEU351 (19 out of 20 peptides) and VAL 354 (20 out of 20 peptides)…”

  1. Explain the significance of interactions observed in the complexes. Interaction analysis could be performed after docking and then it could be further assessed by simulation studies.

Thank you for the suggestion. We agree that in silico interaction analysis would further provide additional info for the studies. However, we will not be able to substantiate the in silico data with an in vitro analysis since we do not possess the capacity to mutate the E protein. Therefore, from the suggestion, we have added these sentences:

Page 7 Line 330: “We believe that these residues might play a crucial role in the effective binding interactions between the peptide fragments and the E protein. However, no further specific investigation (i.e. mutation analysis followed by simulation studies) was instigated.”

  1. The manuscript needs standard review for enriched referencing, especially in the introduction section. For example, from lines 46 to 56 only a single citation is there.

We thank the reviewer for the suggestion. As mentioned in Comment 2, the introduction section was rewritten. More references were added. Please see the manuscript for the changes.

  1. Please do add the future perspective of this study.

We have added the future perspective of this study as suggested. Please see Page 14 Line 634.

Reviewer 2 Report

Dear authors! 

Your conclusions in section 2.3 completely contradict the data in table 1 and figure 5. 

An absolutely obvious candidate for further experimental study is peptide 10-6 with the lowest  AMBER energies (twice as compared to the rest) (see Table 1) and the lowest RMSD (almost twice) (see Fig. 5) In your conclusions, you rely heavily on the energies received by the program of Vina and not on the energies of  AMBER. However, the Vina energies are much more approximate and serve only for ranking purposes. In addition, the energy difference of 0.5 kcal/mol should not be taken into account, since this is the energy value of temperature fluctuations.

Author Response

Reply to Reviewers

Dear reviewers,

 I thank you for the feedback to better improve our manuscript. Please find our answers below. All referencing (Page number and Lines) is referring to the latest manuscript with track changes.

Thank you.

Yours Sincerely,

Abdullah Al-Hadi Ahmad Fuaad

Corresponding Author 1

Comments and Suggestions for Authors from Reviewer 2

  1. Your conclusions in section 2.3 completely contradict the data in table 1 and figure 5.

Thank you for pointing it out. We have corrected our sentences to better explain our thoughts. Please see Page 8 Line 374.

  1. An absolutely obvious candidate for further experimental study is peptide 10-6 with the lowest AMBER energies (twice as compared to the rest) (see Table 1) and the lowest RMSD (almost twice) (see Fig. 5) In your conclusions, you rely heavily on the energies received by the program of Vina and not on the energies of AMBER. However, the Vina energies are much more approximate and serve only for ranking purposes. In addition, the energy difference of 0.5 kcal/mol should not be taken into account, since this is the energy value of temperature fluctuations.

We thank you for the feedback. We do, to a certain degree, agree that DN58opt_10-6 can also be a good candidate. We mentioned about our interest with this peptide on Page 4 Line 259, “… DN58opt_9-9 with a binding affinity of -8.4 kcal/mol and DN58opt_10-6 with a binding affinity of -8.4 kcal/mol could also be potential peptide inhibitors…” However, based on Figure 7, there are red colored residues in DN58opt_10-6 interactions diagram, which indicate potential unfavorable bump between residues. Taking into consideration of the overall picture from our data, including the synthesis and future work (e.g., animal experiments), we have decided to proceed with DN58opt_8-13.

 We have updated Figure 7 to highlight this unfavorable interaction and added an explanation in Section 2.3 as below:

Page 9 Line 381: “...Although DN58opt_10-6 looks promising as a candidate, we have decided not to pursue with synthesis primarily due to the potential unfavorable bump between residues its residues as indicated from Figure 7 (a)…”

Reviewer 3 Report

In the present manuscript “Hit-to-Lead Short Peptides against Dengue Type 2 Envelope protein: Computational and Experimental Investigations”, authors employed docking and molecular simulation to screen a virtual peptide library containing peptide fragments from known DENV 2 peptide inhibitors. They have obtained an 8-amino-acid short peptide which showed good affinity in docking and also demonstrated great inhibition effect in vitro. Overall, I think that the manuscript is well-written, well-structured and would be interested to the readers of molecules. I suggest the authors to further explain a few points and add some more data to better support the conclusion if possible.

  1. Since the standard DET4 has the highest binding affinity among all peptides, could the authors discuss why it demonstrates low inhibition effect in vitro?
  2. Both peptide selected are derived from DN58opt. I am wondering if the authors include the full length DN58opt in the docking, molecular simulation and inhibition assay? What is the result? It will be more convincing if the authors include the full length peptide as a control.
  3. What type of docking have been carried out (rigid, flexible, induced fit?)
  4. For the molecular simulation, is it done in repeats? It seems it is not specified in the manuscript.

Author Response

Reply to Reviewers

Dear reviewers,

 I thank you for the feedback to better improve our manuscript. Please find our answers below. All referencing (Page number and Lines) is referring to the latest manuscript with track changes.

 Thank you.

Yours Sincerely,

Abdullah Al-Hadi Ahmad Fuaad

Corresponding Author 1

Comments and Suggestions for Authors from Reviewer 3

In the present manuscript “Hit-to-Lead Short Peptides against Dengue Type 2 Envelope protein: Computational and Experimental Investigations”, authors employed docking and molecular simulation to screen a virtual peptide library containing peptide fragments from known DENV 2 peptide inhibitors. They have obtained an 8-amino-acid short peptide which showed good affinity in docking and also demonstrated great inhibition effect in vitro. Overall, I think that the manuscript is well-written, well-structured and would be interested to the readers of molecules. I suggest the authors to further explain a few points and add some more data to better support the conclusion if possible.

  1. Since the standard DET4 has the highest binding affinity among all peptides, could the authors discuss why it demonstrates low inhibition effect in vitro?

We thank the reviewer for the comment. Yes, it is a general observation that a molecule with a high binding affinity would therefore has the lowest (least) inhibition potential. In this case, there is no global peptide “standard” for dengue inhibition studies. Therefore, since DET4 was reported as a dengue peptide-based inhibitor, we have decided to use DET4 as our “standard” peptide.

We have additionally added an explanation on Page 9 Line 409: “…Since there is no global peptide “standard” for DENV plaque formation assay, we have selected DET4, a previously reported peptide inhibitor, as our peptide standard. Additionally, we have included a non-peptide viricidal, ribavirin, as the positive control…”

  1. Both peptide selected are derived from DN58opt. I am wondering if the authors include the full length DN58opt in the docking, molecular simulation and inhibition assay? What is the result? It will be more convincing if the authors include the full length peptide as a control.

We thank the reviewer for the suggestions. We did not add these peptides in this study since their potentials were reported previously and that our goal is to study the minimal potential inhibitory sequence of such peptide. Having said that, we sincerely believe that, DET4, which was previously studied in in vitro assay is the best choice for activity comparison.

 Since this work is also a continuation of our previous works, we would like to invite the reviewer to our other published works:

  • Computational identification of self-inhibitory peptides from envelope proteins (2012) – https://doi.org/10.1002/prot.24105
  • Dengue envelope domain III-peptide binding analysis via tryptophan fluorescence quenching assay (2014) - https://doi.org/10.1248/cpb.c14-00165
  • Conformational and Energy Evaluations of Novel Peptides Binding to Dengue Virus Envelope Protein (2017) - https://doi.org/10.1016/j.jmgm.2017.03.010
  • Computational-aided design: minimal peptide sequence to block dengue virus transmission into cells (2020) - https://doi.org/10.1080/07391102.2020.1866074

  1. What type of docking have been carried out (rigid, flexible, induced fit?)

Thank you for your time reviewing our manuscript. We have briefly describe about our approach under Section 3.1.2, where the receptor (E protein) was set to be fully rigid which the peptides were set to be fully flexible, except for the amide bond. This is to ensure best possible docking scenario is achieved. We hope this answers your question.

  1. For the molecular simulation, is it done in repeats? It seems it is not specified in the manuscript.

Thank you for the query. No, it was not done in repeats. We believe that the currently standard of simulation is a minimum of 50 ns, whilst we completed our dynamics studies in 100 ns simulations. Having a more resource intensive run, we hope that the data obtained can provide enough justification that would further substantiate our overall finding.

Reviewer 4 Report

In this paper, the authors reported the research process of DN58opt_8-13, a lead compound as Dengue virus type 2 (DENV2) inhibitors, based on the reported peptides DN58wt, DN58opt, DS36wt, and DS36opt. The research is a complete and in-depth work.  

However, there is some information needed to supplement:

  1. What is the reason that the RMSD values of E protein (orange) in figures 2-5 are with big difference? Especially the values in figure 2 and in figure 4.
  2. The author used Ribavirin as a positive control. What is the action mechanism of Ribavirin? That means, does this drug affect the fusion process of virus into host? Why is the dose of Ribavirin 100 μM, while the other compounds 200 μM?
  3. The authors should provide the copies of all spectra of the synthesized peptides in Supporting Information.

Author Response

Reply to Reviewers

Dear reviewers,

 I thank you for the feedback to better improve our manuscript. Please find our answers below. All referencing (Page number and Lines) is referring to the latest manuscript with track changes.

 Thank you.

Yours Sincerely,

Abdullah Al-Hadi Ahmad Fuaad

Corresponding Author 1

Comments and Suggestions for Authors from Reviewer 4

In this paper, the authors reported the research process of DN58opt_8-13, a lead compound as Dengue virus type 2 (DENV2) inhibitors, based on the reported peptides DN58wt, DN58opt, DS36wt, and DS36opt. The research is a complete and in-depth work.  

However, there is some information needed to supplement:

  1. What is the reason that the RMSD values of E protein (orange) in figures 2-5 are with big difference? Especially the values in figure 2 and in figure 4.

We thank you for the query. The differences are due to the plotted data for each run is different, i.e., each graph illustrates RMSD value following an equilibrated ligand-receptor complex stimulated system, as described on under Section 3.2. As such, each ligand-receptor pair have had its own plot and thus will not be similar.

  1. The author used Ribavirin as a positive control. What is the action mechanism of Ribavirin? That means, does this drug affect the fusion process of virus into host? Why is the dose of Ribavirin 100 μM, while the other compounds 200 μM?

We thank the reviewer for the question.

Ribavirin is a broad-spectrum viricidal and was reported to increase viral mutagenesis that would result in inability for viruses to replicate (https://doi.org/10.1099/vir.0.81655-0). Although the mode of action is not specific to this study intended purposes, we also observed previous reports using ribavirin as the positive control for this specific assay (https://doi.org/10.1371/journal.pone.0126360, https://doi.org/10.1155/2017/1827341, https://doi.org/10.1007/978-1-4939-3387-7_36, to name a few). Moreover, there is no marketed drug against DENV having this specific mode of action (inhibition of fusion process). We believe our use of ribavirin is justified. We have therefore added “…Although ribavirin’s mechanism of action is not via inhibition of DENV fusion process, we observed that ribavirin is the positive control of choice for this type of assay…” on Page 9 Line 412. We hope this answers the questions.

  1. The authors should provide the copies of all spectra of the synthesized peptides in Supporting Information.

Spectra will be added as requested.

Round 2

Reviewer 1 Report

Accepted for publication.

Author Response

Dear reviewer,

Thank you for your time.

Reviewer 2 Report

The authors talk about a large adverse contribution to binding, but the binding value itself is large. Additional lengthy explanations are needed. What part of the unfavorable binding in the overall picture of binding is given by the amino acid residues shown in Figure 7 (a)

Author Response

Dear Reviewer,

Thank you for your feedback. Please find our answer to your comments below.

Answer to Comments/Suggestions:

Q: The authors talk about a large adverse contribution to binding, but the binding value itself is large. Additional lengthy explanations are needed. What part of the unfavourable binding in the overall picture of binding is given by the amino acid residues shown in Figure 7 (a)

We thank you for your comments and have relooked at the relevant section. We have removed our previous answer (Page 9 Line 269) and replaced it with a more detailed explanation. We have also relooked at the interaction and have supplemented our findings with a Supplementary Data - Figure S1 (please visit https://bit.ly/36lXi4b to view the figure). Please find our answers below. We hope this answers your queries. 

From:

Page 9, Line 269: "... Although DN58opt_10-6 looks promising as a candidate, we have decided not to pursue with synthesis primarily due to the potential unfavorable bump between its residues as indicated from Figure 7 (a) – denoted by the red dots..."

To:

Page 9, Line 269: "...Despite the predicted strong vina affinity, MMGBSA and MMPBSA energies of DN58opt_10-6, it was not selected for further synthesized and biological assay work because there is a potential unfavorable positive-positive repulsion between Lys36 residue of E protein with the histidine residue (HIS6: 6th amino acid) of DN58opt_10-6, as well as potential unfavorable donor-donor repulsions between ARG350 residue of E protein with the tyrosine and cysteine residues (TYR1 and CYS3: 1st and 3rd amino acids, respectively) of DN58opt_10-6 (please refer to supplementary data, Figure S1, at https://bit.ly/36lXi4b). These repulsive interactions would pose a risk for DN58opt_10-6 to not be able to stay at the binding site stably in the biological assay, especially the predicted MD stability is only in a nanosecond scale..."

Reviewer 3 Report

I think all questions have been answered by the authors and the revised version is suitable for publication.

Author Response

The reviewer,

Thank you for your time.